# Stroke-Induced Secondary Neurodegeneration of the Corticospinal Tract—Time Course and Mechanisms Underlying Signal Changes in Conventional and Advanced Magnetic Resonance Imaging

**DOI:** 10.3390/jcm13071969

**Published:** 2024-03-28

**Authors:** Marialuisa Zedde, Ilaria Grisendi, Federica Assenza, Manuela Napoli, Claudio Moratti, Giovanna Di Cecco, Serena D’Aniello, Franco Valzania, Rosario Pascarella

**Affiliations:** 1Neurology Unit, Stroke Unit, Azienda Unità Sanitaria Locale-IRCCS di Reggio Emilia, Viale Risorgimento 80, 42123 Reggio Emilia, Italy; grisendi.ilaria@ausl.re.it (I.G.); federica.assenza@ausl.re.it (F.A.); valzania.franco@ausl.re.it (F.V.); 2Neuroradiology Unit, Azienda Unità Sanitaria Locale-IRCCS di Reggio Emilia, Viale Risorgimento 80, 42123 Reggio Emilia, Italy; napoli.manuela@ausl.re.it (M.N.); moratti.claudio@ausl.re.it (C.M.); giovanna.dicecco@ausl.re.it (G.D.C.); daniello.serena@ausl.re.it (S.D.); pascarella.rosario@ausl.re.it (R.P.)

**Keywords:** secondary neurodegeneration, SND, Wallerian degeneration, stroke, magnetic resonance imaging, MRI, diffusion tensor imaging, DTI

## Abstract

Secondary neurodegeneration refers to the final result of several simultaneous and sequential mechanisms leading to the loss of substance and function in brain regions connected to the site of a primary injury. Stroke is one of the most frequent primary injuries. Among the subtypes of post-stroke secondary neurodegeneration, axonal degeneration of the corticospinal tract, also known as Wallerian degeneration, is the most known, and it directly impacts motor functions, which is crucial for the motor outcome. The timing of its appearance in imaging studies is usually considered late (over 4 weeks), but some diffusion-based magnetic resonance imaging (MRI) techniques, as diffusion tensor imaging (DTI), might show alterations as early as within 7 days from the stroke. The different sequential pathological stages of secondary neurodegeneration provide an interpretation of the signal changes seen by MRI in accordance with the underlying mechanisms of axonal necrosis and repair. Depending on the employed MRI technique and on the timing of imaging, different rates and thresholds of Wallerian degeneration have been provided in the literature. In fact, three main pathological stages of Wallerian degeneration are recognizable—acute, subacute and chronic—and MRI might show different changes: respectively, hyperintensity on T2-weighted sequences with corresponding diffusion restriction (14–20 days after the injury), followed by transient hypointensity of the tract on T2-weighted sequences, and by hyperintensity and atrophy of the tract on T2-weighted sequences. This is the main reason why this review is focused on MRI signal changes underlying Wallerian degeneration. The identification of secondary neurodegeneration, and in particular Wallerian degeneration, has been proposed as a prognostic indicator for motor outcome after stroke. In this review, the main mechanisms and neuroimaging features of Wallerian degeneration in adults are addressed, focusing on the time and mechanisms of tissue damage underlying the signal changes in MRI.

## 1. Introduction

Neurodegeneration is a general term summarizing the results of several pathophysiological processes leading to cell death and loss of substance within the central nervous system (CNS). The neurodegeneration might be a primitive phenomenon or a secondary consequence of a primary brain injury. In this review, we refer to neurodegeneration in a physiological sense and not to describe a definite category of neurological disease that is mainly underlined by primary neurodegeneration. This last one is called secondary neurodegeneration, and it refers to the loss of substance and function in brain regions connected to the site of a primary injury. These areas might be near or remote from the site of the primary injury. Several studies have addressed the process of secondary neurodegeneration in neurological diseases (stroke, trauma or surgical intervention, neoplasms, demyelinating diseases, etc.) [1]. As for other processes in the brain, the timing of secondary neurodegeneration involves different mechanisms. Indeed, the acute stage (5–12 weeks after the primary injury) is mainly due to excitotoxic mechanisms and oxidative stress, producing cytotoxic edema. The chronic phase (>12 weeks after the primary injury) is characterized by cellular death, gliosis, and end-stage atrophy [2,3]. The main imaging techniques for identifying secondary neurodegenerations are based on magnetic resonance imaging (MRI), and in particular on diffusion-based MRI techniques, namely diffusion weighted imaging (DWI)-MRI and diffusion tensor imaging (DTI)-MRI. In the acute phase, axonal degeneration might exhibit a hyperintense signal on DWI-MRI with a corresponding hypointense signal on the apparent diffusion coefficient (ADC) map because of edema and cellular vacuolization [4]. DTI-MRI is a technique that uses different gradient directions to generate images based on the random motion of water molecules. It is the most sensitive technique for depicting microstructural changes in brain white matter. Secondary neurodegeneration shows a reduction in fractional anisotropy secondary to axonal and myelin sheath damage.

In this review, the main neuroimaging features of stroke-induced Wallerian degeneration of the corticospinal tract in adults are illustrated together with the underlying pathomechanisms and the time course of signal changes in order to provide a practical interpretation of a common and neglected finding.

## 2. Definitions

Secondary neurodegeneration can be separated into two types: (I) axonal degeneration and (II) transneuronal degeneration [5]. Axonal degeneration refers to the loss of integrity by axonal fragmentation and loss of function of the axons after an injury of the neuronal cell body or in the proximal part of the axon itself. Axonal degeneration might be anterograde and retrograde, as detailed in Table 1. Anterograde axonal degeneration, also called Wallerian degeneration, is the most commonly reported type and refers to the degeneration of the axon distal to an injury. If the injury involves the neuronal cell body (e.g., because of a lesion of pyramidal cells in the motor cortex), only anterograde axonal degeneration occurs. If the injury involves the axon along its course (e.g., because of a lesion in the internal capsule), both anterograde and retrograde axonal degeneration occur, with the latter referring to the progressive disruption of the axon proximal to the site of injury. Both subtypes of axonal degeneration do not involve other neurons beyond the synaptic connection [5].

Conversely, transaxonal degeneration, also called trans-synaptic or transneuronal degeneration, involves more than one neuron in a pathway, and it refers to afferent or efferent neuron degeneration through the synaptic connection, being due to the loss of synaptic output or input [5,7]. Anterograde axonal degeneration in the pyramidal tract, or Wallerian degeneration, is the main focus of our description. Anterograde axonal degeneration, or Wallerian degeneration, was first described in 1850 by Augustus Waller [8] as the distal axonal fragmentation after neuron cell-body or proximal axonal injury. The biochemical and molecular basis for this involves mostly excitatory neurotransmitters and proinflammatory cytokines as triggers of the initial injury [3], leading to microtubule disassembly, axon swelling and fragmentation, and finally waste removal by locally recruited phagocytes [9]. The pyramidal tract and corpus callosum are the most frequently affected pathways of axonal degeneration. The neuron bodies of the pyramidal tract are located in the primary motor cortex (precentral gyrus), the somatosensory cortex (postcentral gyrus), the supplementary motor area (the dorsomedial aspect of the superior frontal gyrus), and the premotor cortex (dorsal frontal superior and middle gyri). The axons originating from these neurons form two tracts, the corticobulbar and corticospinal tract, which together constitute the pyramidal tract, and this travels through the superior corona radiata, posterior limb of the internal capsule, cerebral peduncle, basis pontis, and the anterior portion of the medulla oblongata, where it forms the medullary pyramids. At the lower limit of the medullary pyramids, approximately 75–90% of the pyramidal tract decussates (pyramidal decussation of Mistichelli), becoming the lateral corticospinal tract, and the remaining 10% runs along the ipsilateral spinal cord, forming the anterior corticospinal tract [10,11]. An injury occurring in any segment of the primary motor cortex and/or pyramidal tract can provoke a Wallerian degeneration. Ischemic stroke in the middle cerebral artery (MCA) territory is the most frequent cause of this phenomenon.

## 3. Stroke-Induced Secondary Neurodegeneration of the Corticospinal Tract

Stroke-induced Wallerian degeneration is a well-known phenomenon, and it has been described after both ischemic and hemorrhagic stroke, but the former has been more frequently reported in the literature. After ischemic infarction of the motor cortex or descending motor pathways, Wallerian degeneration of the pyramidal tract is the consequence of severe damage in this tract. In past studies, it has been postulated that the evidence of severe pyramidal damage in the early phase after stroke could help to predict the prognosis [12]. In a retrospective and prospective study involving respectively 520 patients with various intracranial lesions and 18 stroke patients [1], Wallerian degeneration was documented in 31/520 and in 10/18 patients, respectively. In the retrospective group, this number comprised 14/220 ischemic stroke and 4/147 hemorrhagic stroke patients. In the prospective group, 6/10 patients with ischemic stroke and 4/8 patients with hemorrhagic stroke investigated 4 months after the stroke were included. Among the patients with ischemic stroke, four had whole MCA territory infarction and two had involvement of the posterior limb of the internal capsule. Among the four hemorrhagic patients, three had involvement of the basal ganglia and centrum semiovale, and one had involvement of the fronto-parietal lobe. A single institution study [12] enrolled nine consecutive patients with first-ever supratentorial ischemic stroke involving the motor cortex and/or the corticospinal tract and performed DTI within the first 16 days of symptom onset. Most patients suffered from striatocapsular infarction (*n* = 4) or lacunar stroke (*n* = 3) affecting the internal capsula. Two patients presented with territorial infarction of the peripheral MCA branches, including the motor cortex. The study showed early signs of Wallerian degeneration in all investigated patients. Pujol et al. [13] demonstrated Wallerian degeneration in the pyramidal tract in 78.6% cases of capsular infarct and found a good correlation with functional deficits in these patients, as did Orita et al. [14], thus concluding that the extent of Wallerian degeneration is related to the severity of motor deficit, as suggested by other studies using either MRI or transcranial magnetic stimulation [15,16,17,18].

## 4. MRI Features and Timeline of Stroke-Induced Secondary Neurodegeneration

In general, Wallerian degeneration is characterized by a highly stereotypical course, sequentially characterized by the disintegration of axonal structures (within days after injury), degradation of myelin because of the infiltration of macrophages, and fibrosis and atrophy of the affected fiber tracts [19,20,21]. Conventional MRI detects signal intensity changes that vary during the time course of Wallerian degeneration [1,14,16,17,22,23,24,25], but these signal intensity changes are generally not detected until 4 weeks after stroke. After a short phase of hypointensity in T2W-MRI [1,23], a T2-WI hyperintensity along the affected tracts is usually found in the subacute and chronic stage that is most likely due to fibrosis of the pyramidal tract [1,14,16,22,23,24,25]. In the early MRI studies, the diagnosis of Wallerian degeneration was made without using DWI sequences [14,23], using instead four distinct time-specific signal intensity changes of Wallerian degeneration staged from I to IV [1] (Table 2).

In addition, Wallerian degeneration is characterized by a DWI hyperintensity along the course of the pyramidal tract, mainly visible in the internal capsule and brainstem, at the early subacute stage of stroke. This appearance sometimes might lead to a misdiagnosis of these hyperintensities as new ischemic lesions. The suggested mechanisms underlying the MRI signal changes in Wallerian degeneration after ischemic or hemorrhagic stroke include several issues: increased water content, glial proliferation, and compartmentalization of the extracellular and intracellular water protons [23]. The signal changes in DWI in acute ischemia are thought to be related to increased cell-membrane permeability and cytotoxic edema [26]. However, signal intensity changes in DWI are not able to discriminate between the fluid shifts caused by ischemia and those arising secondary to the effects of neuronal dysfunction [27]. The evolution of signal-intensity changes reflects typical stereotyped biochemical changes within the white-matter tract that occur during Wallerian degeneration, which essentially consist of stages of axonal breakdown, myelin breakdown, and gliosis [23,28]. The time course of Wallerian degeneration as described in the corticospinal tract encompasses three phases (acute, subacute, and chronic), and the main MRI features are summarized in Table 3.

The hypointensity in T2WI in the subacute phase is mainly due to an alteration in protein–lipid–water content as a result of degradation and disintegration of the myelin sheath [29]. In the chronic stage of Wallerian degeneration, atrophy of the pyramidal tract can be detected by computed tomography, as documented in early studies [30]. On the contrary, transaxonal neurodegeneration (e.g., hypertrophic olivary degeneration) lacks DWI hyperintensity in any temporal phase of its evolution, with an early T2WI hyperintensity of the inferior olivary nucleus [31,32]. However, MRI does not show a lesion in 44% of cases of hypertrophic olivary degeneration [31]. An example of the signal changes in Wallerian degeneration in conventional MRI is provided in Figure 1.

In the first days and weeks, Wallerian degeneration is not detectable by conventional MRI, but some information may be provided within this time-frame by advanced MRI techniques. One of the DWI-based techniques providing information on secondary neurodegeneration is DTI [33], because of its capability of demonstrating in vivo the axonal architecture and connectivity. Indeed, although white matter is well imaged on the T1, T2, and FLAIR sequences, which are routinely used for conventional MRI, DTI is able to map, voxel by voxel, the fiber direction and the degree of anisotropy due to the axonal membranes and myelin sheaths limiting the random motion of the water molecules. DTI is performed by applying diffusion sensitizing gradients in at least six non-collinear directions, aiming to determine the full diffusion tensor [34]. The diffusion tensor provides information on the predominant direction and degree of water diffusion. Then, it gives clues on the microstructural organization of the tissue [35]. The water diffusion in white matter is expected to be fast along the main fiber directions and slow perpendicular to the fibers’ course. This feature is registered as anisotropic diffusion, and its degree is strictly dependent on the level of organization of the bundles. In fact, the orientation of axonal membranes and myelin sheaths affects the degree of obstacle to water diffusion. These features make DTI an adequate technique to investigate the Wallerian degeneration of long descending tracts after ischemic stroke. In more detail, the tensor matrix is an ellipsoid, and its principal axis is oriented in the direction of maximum diffusivity. Through matrix diagonalization, three eigenvalues are obtained that represent the apparent diffusivity in the three main axes of the ellipsoid, named the major (λ_1_), medium (λ_2_), and minor (λ_3_) axes or eigenvectors. Fractional anisotropy (FA) provides a measurement of the diffusivity in a predefined coordinate system and ranges from 0 (isotropy) to 1 (maximum anisotropy). In addition, DTI might discover variations in the anatomy of corticospinal tracts, identifying the fibers that do not cross over to the ipsilateral lateral corticospinal tract and the ones decussating to constitute the contralateral anterior corticospinal tract [36,37]. In fact, the corticospinal tracts might be mildly asymmetric, and the left one has higher FA values and lower transverse diffusivity; this phenomenon is mainly due to its higher myelin content than the contralateral one [38]. DTI can identify Wallerian degeneration early after stroke [12], although its first application was in the chronic phase of stroke [39,40], and reduced anisotropy of the pyramidal tract of the affected side was considered the imaging equivalent of Wallerian degeneration. This last one is primarily a pathological term. A single institution study [12] used DTI within 16 days from stroke onset to study early Wallerian degeneration of the pyramidal tract after acute ischemic stroke, finding a reduced anisotropy in FA maps and typical changes in eigenvalues of the cerebral peduncle of the affected side: the first eigenvalue is decreased, and the third eigenvalue is increased. T2-WI and averaged ADC maps, acquired at the same time, did not reveal any changes. The extent of Wallerian degeneration, measured by decreased FA, correlated with the motor deficit both at the time of the DTI study and after 3 months. A potential bias in studies comparing DTI metrics in the affected and contralateral side in the same regions of interest (ROIs) is that they may consider healthy the contralateral white matter after stroke. In fact, a study reported that the FA of the normal-appearing white matter was increased in the whole brain within 2 years after stroke, suggesting a long-term improvement in apparent white-matter integrity [41]. Moreover, comparing the rate of FA reduction in studies conducted at different times after stroke, a progressive reduction of FA seems to be seen over months. Thomalla et al. [12] documented a 13% reduction in mean FA in the cerebral peduncle on the affected side 2–16 days after stroke. Werring et al. [40] found a 15% decrease in FA in the cerebral peduncle, 2–6 months after stroke. Pierpaoli et al. [39] found a 32% decrease in FA in the cerebral peduncle more than one year after stroke. Histopathological and experimental data provided some clues about how water diffusion might change in fiber tracts during the early stages of Wallerian degeneration. In particular, histopathological studies have shown a highly stereotyped time course of changes. In the first stage, which occurred within days, the main findings are the disintegration of axonal structures and fragmentation of the myelin sheaths [19,20,21], and the time frame of histological detection of degeneration in descending axons was as early as 2–7 days after the insult in rats [42]. In the second stage, starting at about 2 weeks after the injury, the main findings are the degradation of myelin and infiltration of astrocytes [18,19,20,21], leading to the loss of axonal structures, which results in less restricted diffusion perpendicular to the main direction of the fibers (elevated second and third eigenvalues). Membrane disintegration and cellular debris create new barriers to the free diffusion of water molecules, and at this stage, the diffusivity decreases parallel to the main fiber direction (leading to a reduced first eigenvalue). The combination of these effects should produce reduced anisotropy, but the effect on diffusion indices based on axon orientation is not easy to predict. Moreover, the average diffusivity might remain unchanged. The time course of signal changes in DTI in Wallerian degeneration of the corticospinal tract after stroke was investigated in a study [43] of nine Asian patients vs. matched controls with motor-pathway subcortical infarction and negative findings outside the infarction by conventional MRI. All enrolled subjects underwent DTI at 1 week, 2 weeks, 1 month, 3 months, and 12 months from stroke, and the following dynamic changes in the degenerated corticospinal tract were identified: (I) the relative FA (rFA) of the degenerated corticospinal tract decreased during the first 3 months, and then remained relatively unchanged; (II) the relative mean diffusivity (MD) was stable during the first 2 weeks, and gradually increased until 3 months, reaching a new relatively stable level. Interestingly, no correlation was found between changes in the rFA of the corticospinal tract within the first 2 weeks and the motor outcome at 1 year, suggesting that early changes of the rFA of the corticospinal tracts could be used to predict motor outcomes in stroke patients. In addition, this study allows some considerations about the pathological processes underlying the signal changes in MRI. Indeed, Wallerian degeneration in the CNS is characterized by a series of subsequent and simultaneous phenomena lasting from several hours to days after the time of the lesion (Table 4):

All these processes affect the MRI signal and, in particular, the diffusion tensor eigenvalues. DTI is able to see the early phase of Wallerian degeneration when it is performed after a median delay of 9 days [12] but not when it is performed within the first week [43] after stroke. This finding suggests that the pathologic changes of the affected corticospinal tract are not apparent enough to be detected by DTI within 1 week after stroke. Liang et al. [48] longitudinally assessed both anterograde and retrograde Wallerian degeneration using DTI in 12 patients with subcortical ischemic infarctions involving the internal capsule in its posterior limb, finding a progressively decreased FA from the first week located in the region just proximal and distal to the internal capsule lesion. Yu et al. [43] found changes in diffusion measurements at the second DTI at 2 weeks. Similarly, in a longitudinal study [49] involving 60 patients with MCA stroke investigated by using MRI with DTI within 12 h of symptom onset, and then on days 3 and 30, only the last DTI identified changes in the diffusivity measurements. In more detail, Yu et al. [43] found a sharply decreased FA, a reduced λ_1_ eigenvalue, an increased λ_23_, and an unchanged MD at 2 weeks. The decrease in λ_1_ was probably due to the fragmentation of axons and the consequent barrier to the longitudinal diffusion of water [45]. Indeed, the increase in λ_23_ was attributed to the degradation of the myelin sheaths, making water more mobile perpendicular to the axons. A similar conclusion was supported by an experimental study performed on a mouse model of retinal ischemia [50]. The value of λ_1_ did not decrease further from 14 to 30 days after stroke [43], probably because of the complete fragmentation of the axons. However, λ_23_ increased further, and the cause of this phenomenon was probably the continuing degradation of the myelin sheaths with the clearance of the axonal and myelin debris (see Table 4). The final results of the changes in the diffusion tensor eigenvalues are a decrease in FA and an increase in MD. The λ_1_ value slightly increases from 30 to 90 days after stroke, because most of the axonal fragments are cleared and water might diffuse in the longitudinal direction [43]. Conversely, λ_23_ increases because of the clearance of the axonal and myelin debris and new myelin formation without axons. The decrease in FA and the increase in MD are the expected results of the changes in the diffusion tensor eigenvalues. From the 4th month to 1 year after stroke, none of the diffusion indices showed significant changes, suggesting that Wallerian degeneration is relatively stable 3 months after stroke. Puig et al. [49] tried to establish an rFA of 0.925 as a cutoff value for Wallerian degeneration at 30 days using DTI. It is a difficult task, because changes in diffusion after Wallerian degeneration strongly depend on the pre-existing white-matter architecture [39]. Experimental studies suggest that the orientation of membranes is the major determinant of anisotropy after Wallerian degeneration. In fact, changes in diffusion in the rostral pons, at the level where the corticospinal tract intersects the transverse pontine fibers, are substantially different from the ones found in the cerebral peduncle or in the posterior limb of the internal capsule, where isolated and well-defined bundles of parallel fibers constitute the corticospinal tract [51]. Consequently, after white-matter damage, cutoff anisotropic values might be different at different points along the corticospinal tract, without a conclusive result until now. A different approach using ADC maps was proposed by De Vetten et al. [52] in a small cohort of 20 patients with ischemic stroke in the MCA territory. The MRI investigation was performed with a 3.0 T scanner at the following five time points: (1) baseline (≤6 h); (2) 12 + 6 h; (3) 24 + 6 h; (4) 7 + 2 days; and (5) 30 + 10 days. DWI, ADC maps, and FLAIR images were analyzed. The main focus of the analysis was the corticospinal tract at the level of the cerebral peduncle. Decreases in the ADC were evident within 12 h in poor-outcome patients, as previously documented for neonatal and pediatric stroke [53,54]. That descending corticospinal tract ADC values were maximally decreased at 7 days and returned toward baseline by 30 days is consistent with the well-described evolution of diffusion signals in hemispheric infarcts [55,56]. Interestingly, MR signal evolution in brain-stem infarcts has been described as delayed [57]. The protracted time course of descending corticospinal tract diffusion changes is consistent with an evolving pathological process, such as Wallerian degeneration, although acute pathological studies confirming this are required.

Finally, few studies specifically addressed the time course of Wallerian degeneration in hemorrhagic stroke. One of these [58] included 27 patients with spontaneous supratentorial hemorrhage, on whom serial MRIs were performed at 2, 7, 14, and 21 days. Among these, thirteen patients (48%) exhibited Wallerian degeneration of the corticospinal tract, with a significant prevalence of deep hemorrhages. Wallerian degeneration of the corticospinal tract was first appreciated as a hyperintense DWI signal and by a corresponding hypointensity on ADC maps. The reduced diffusion on ADC maps lasted for a median of 11 days (IQR, 8 to 12 days) before becoming hyperintense. T2/FLAIR images showed hyperintensity in the same areas at a median of 11 days (IQR: 6 to 14 days) in 12 of 13 patients.

## 5. Prognostic Information

The presence of Wallerian degeneration in the CNS means an irreversible loss of neuronal function because there is little evidence of axonal regeneration in the CNS. Early identification of this phenomenon could be useful in predicting the motor outcome. By correlating DTI findings and clinical data, Puig et al. [49] observed that motor deficits at 30 days after stroke are inversely related to the anisotropy in the pons. The decrease in FA values in the affected corticospinal tract progressively reduces during the subacute-to-chronic stages of stroke, and previous studies have demonstrated worse outcomes in patients with evident Wallerian degeneration by conventional MRI [1,23,24,58,59,60]. Abnormalities in signal intensity related to Wallerian degeneration are generally not detectable until 4 weeks after stroke. At this time, the main finding is a hyperintensity in DWI and T2W images along the affected tracts. Moreover, some patients with motor deficits have differences in anisotropy but no signal-intensity abnormalities in the corticospinal tract by conventional MRI, and this subgroup is the most prevalent at 30 days [49]. In addition, greater FA reduction along the affected corticospinal tract after stroke is associated with greater motor deficit in the early phase and worse motor recovery at 3 months [12,61,62,63]. Wanatabe et al. [18] assessed Wallerian degeneration in the corticospinal tract in 16 patients with stroke (6 ischemic, 10 hemorrhagic), using serial DTI evaluations to measure FA in the rostral pons. The good-recovery group had no significant change in anisotropy, whereas the poor-recovery group had a significant decrease in anisotropy between 2 and 3 weeks. In a small subgroup of patients with intracerebral hemorrhage, FA indices measured in the cerebral peduncles within 2 days after intracerebral hemorrhage onset may predict the functional motor outcome [64], with a cutoff point 0.85 of the rFA for the good (m-NIHSS, 0–2) and poor (m-NIHSS scores ≥ 3) outcomes for the motor items of the National Institute of Health Stroke Scale.

## 6. Conclusions

Secondary neurodegeneration in the CNS is a relatively neglected and poorly investigated phenomenon. In particular, Wallerian degeneration of the corticospinal tract after stroke shows a delayed and remote replication of the same pathological stages of tissue damage in the brain regions affected by stroke. A timely diagnosis in the early phase could help in designing neuroprotective trials and provide some prognostic information on motor outcomes. Several MRI techniques might be used to characterize this phenomenon and its course, but each of these techniques has several limitations. T2W-MRI identifies Wallerian degeneration when the tract is completely disrupted after weeks from the injury. DWI-MRI allows for an early identification of Wallerian degeneration, but it takes a few weeks to become apparent. Finally, DTI-MRI could allow an early identification of Wallerian degeneration, but at the moment, it provides relevant pathophysiological information without a quantitative threshold, helping to diagnose this phenomenon in individual patients in clinical practice. In fact, there are not agreed and demonstrated thresholds to use for these purposes in conventional MRI and DTI metrics, and further studies are needed to assess this phenomenon.

## Figures and Tables

**Figure 1 jcm-13-01969-f001:**
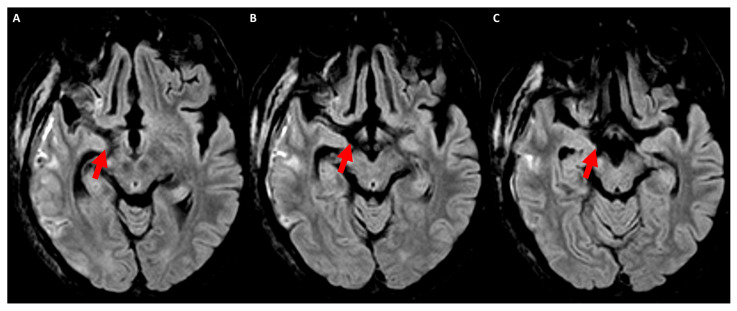
Axial FLAIR MRI acquired at 4 weeks after ischemic stroke in the right middle cerebral artery territory because of endocarditis involving the posterior limb of the internal capsule within the territory of the left anterior choroidal artery. Panels (**A**–**C**) show sequential slices from top to bottom at the level of the cerebral peduncle. The hyperintense signal (red arrows) corresponding to the course of the corticospinal tract is clearly evident on the right side in comparison with the normal signal on the left side.

**Table 1 jcm-13-01969-t001:** Main features of anterograde and retrograde axonal degeneration [6].

Subtype of Axonal Degeneration	Definition	Main Mechanisms
Anterograde	Axonal degeneration secondary to an injury to the neuronal cell body or proximal axon	Axonal fragmentation Macrophage infiltration Degradation of the myelin sheath Fibrosis and atrophy of the tract
Retrograde	Axonal degeneration secondary to an injury in the distal axon	Axonal disintegration Myelin degradation

**Table 2 jcm-13-01969-t002:** Signal intensity stages of Wallerian degeneration in early MRI studies [5,13,14,17].

Stage	MRI Signal Features
I	no signal intensity abnormality
II	hypointense signal on T2 and proton density (PD) weighted images
III	hyperintense signal on T2 weighted images
IV	brainstem atrophy

**Table 3 jcm-13-01969-t003:** Time course of Wallerian degeneration with main MRI features [5,12,13,14,17,18,25].

Temporal Phases	DWI	ADC	T2W/FLAIR
Acute	Restricted diffusion in the CST, distal to the primary injury	ADC values decreased within 12 h, was maximal at 7 days, and normalized after 30 days after the primary injury	Short phase of T2W hypointensity, followed by a hyperintense signal corresponding to DWI hyperintensity starting at approximately 12 days after the primary injury
Subacute	-	-	Hypointense signal along the tract
Chronic	-	-	Hyperintense signal along the tract and atrophy

**Table 4 jcm-13-01969-t004:** Sequential underlying pathological processes in Wallerian degeneration.

Temporal Phases	Processes
Acute	- disintegration of the cytoskeleton of the axons into axonal fragments [44,45] - the myelin sheaths surrounding the axons become less tightly wrapped and discontinuous, breaking into ovoid structures [44] - microglial activation in the degenerating fibers, followed by the appearance of round macrophages, occurs on days 18–21, after an early transient period of microglial activation in the degenerating fibers
Subacute	- the activated microglial cells digest the axonal and myelin debris, with a complete clearance of both axonal and myelin debris from the CNS in 90 days [46]
Chronic	- axonal regeneration and myelin formation do not occur because of the apoptosis of oligodendrocytes in the first few weeks following injury [47]

## Data Availability

Not applicable.

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
