# Peer review of "Stroke-Induced Secondary Neurodegeneration of the Corticospinal Tract—Time Course and Mechanisms Underlying Signal Changes in Conventional and Advanced Magnetic Resonance Imaging"

_jcm, 2024, doi:10.3390/jcm13071969_

Round 1

Reviewer 1 Report

Comments and Suggestions for Authors

Journal: JCM (ISSN 2077-0383)

Manuscript ID: jcm-2906856

Type: Review

Title: Stroke-induced secondary neurodegeneration of the corticospinal tract. Time course and mechanisms underlying signal changes in conventional and advanced Magnetic Resonance Imaging.

Revision suggestions for the article are listed below.

1-Regarding the 'Abstract' section

1. An explanation containing the term 'Time course and mechanisms underlying signal changes in conventional and advanced Magnetic Resonance Imaging' included in the article title should be included in this section.

2. A sentence explaining the term 'Wallerian degeneration' included in this section should be added.

3. In the article prepared as a review, the quantitative characteristics of the studies revised and included in the article should be briefly mentioned.

4. Keywords should be rearranged to include the terms included in the 'abstract' section (including Wallerian degeneration).

2. Regarding the

1. 'Introduction' section

1-If the term 'Neurodegeneration' is used in a pathophysiological sense, a statement explaining this term should be included.

2- The entire article must be reviewed again, spelling and sentence errors must be corrected, and the article must be written within the framework of the journal rules (including sentence indentations).

2. Definitions section

1-In this section, the meaning of each of the degeneration types mentioned should be explained, and then their subclassifications, if any, should be stated. This part should be restructured.

3. MRI features and timeline of stroke-induced secondary neurodegeneration

1- The term 'DTI MRI' is shown as an imaging method on its own in the article introduction and is evaluated within 'DWI-based techniques' in this section.

In this respect, either a separate paragraph should be created for 'DTI MRI' in this section, and the technique should be explained, or 'DTI MRI' should be added to this section by mentioning 'DWI-based techniques' in the 'Introduction' section. There is a technical error in the article in this respect.

2-Spelling and spelling errors need to be corrected.

3-In this section, more current literature information confirming the relevant data should be included, and this section should be shortened by summarizing it with the mentioned technical method.

3. Regarding the 'Conclusion' section

1-The relevance of each technique included in this section to the subject of the article should be concluded. If there is a meaningful technique, it should be included with quantitative results.

Author Response

1-Regarding the 'Abstract' section

  1. An explanation containing the term 'Time course and mechanisms underlying signal changes in conventional and advanced Magnetic Resonance Imaging' included in the article title should be included in this section.

Thanks for this suggestion. We added the following sentence in the abstract: “The different sequential pathological stages of secondary neurodegeneration provide the interpretation of the signal change on MRI according with the underlying mechanisms of axonal necrosis and repair.”

  1. A sentence explaining the term 'Wallerian degeneration' included in this section should be added.

We changed the sentence referring to Wallerian degeneration as follows: “Among the subtypes of post-stroke secondary neurodegeneration, axonal degeneration of the corticospinal tract, also known as Wallerian degeneration, is the most known and it directly involves motor functions, being crucial for the motor outcome”

  1. In the article prepared as a review, the quantitative characteristics of the studies revised and included in the article should be briefly mentioned.

We would like to thank the reviewer for his/her comments. About this issue, we added the following sentence in the abstract: “Depending on the employed MRI technique and on the timing of imaging, different rates and thresholds of Wallerian degeneration has been provided in the literature. This is the main rea-son why this review is focused on MRI signal changes underlying Wallerian degeneration. The main limitation is that a threshold in Diffusion-based MRI techniques for Wallerian degeneration has not been validated in the early stages.”. The main problem is that there is no diagnostic modality with validated and accepted thresholds to diagnose Wallerian degeneration. DTI metrics is really interesting, but each patient is the control for himself/herself during time. The contralateral hemisphere cannot be considered as normal reference (we pointed out this issue in the text) and we have globally chosen to focus attention on the characteristics, timing of appearance and mechanism underlying the signal alteration in MRI, with any technique. These limitations make it difficult to be truly quantitative in the description, if not, as done, describing the main studies with their information content and the difficult comparability between them.

  1. Keywords should be rearranged to include the terms included in the 'abstract' section (including Wallerian degeneration).

Thanks. We added “Wallerian degeneration” among the keywords.

  1. Regarding the

  1. 'Introduction' section

1-If the term 'Neurodegeneration' is used in a pathophysiological sense, a statement explaining this term should be included.

2- The entire article must be reviewed again, spelling and sentence errors must be corrected, and the article must be written within the framework of the journal rules (including sentence indentations).

Thanks! We added this sentence in the introduction: “In this review we refer to neurodegeneration with a physiological meaning and not for describing a definite category of neurological disease, mainly underlined by primary neurodegeneration.”

We reviewed and checked the language, changing some sentences.

  1. Definitions section

1-In this section, the meaning of each of the degeneration types mentioned should be explained, and then their subclassifications, if any, should be stated. This part should be restructured.

Thanks for the suggestion. We added the following sentences:

“Axonal degeneration refers to the loss of integrity by axonal fragmentation, and loss of function, of the axons after an injury of the neuronal cell body or in the proximal part of the axon itself. Axonal degeneration might be anterograde and retrograde, as detailed in table 1. Anterograde axonal degeneration, also called Wallerian degeneration, is the most commonly reported one and refers to the degeneration of the axon distal to an injury. If the injury involves the neuronal cell body only anterograde axonal degeneration occurs (e.g. because of a lesion of pyramidal cells in the motor cortex). If the injury involves the axon along its course (e.g. because of a lesion in the internal capsule) both anterograde and ret-rograde axonal degeneration occur, and the last one refers to the progressive disruption of the axon proximal to the site of injury. . Both subtypes of axonal degeneration do not in-volve other neurons beyond synaptic connection [5].” 

  1. MRI features and timeline of stroke-induced secondary neurodegeneration

1- The term 'DTI MRI' is shown as an imaging method on its own in the article introduction and is evaluated within 'DWI-based techniques' in this section.

 In this respect, either a separate paragraph should be created for 'DTI MRI' in this section, and the technique should be explained, or 'DTI MRI' should be added to this section by mentioning 'DWI-based techniques' in the 'Introduction' section. There is a technical error in the article in this respect.

Thanks for the suggestions. We rearranged the introduction accordingly with the proposal of the reviewer:

“The main imaging techniques to identify secondary neurodegenerations are based on Magnetic Resonance Imaging (MRI) and, in particular on Diffusion-based MRI techniques, namely Diffusion Weighted Imaging (DWI)-MRI and Diffusion Tensor Imaging (DTI)-MRI. In particular, iIn the acute phase axonal degeneration might exhibit an hyperntense sig-nal on Diffusion Weighted Imaging (DWI)-MRI with a corresponding hypointense signal on Apparent Diffusion Coefficient (ADC) map because of edema and cellular vacuolization [4]. Moreover, Diffusion Tensor Imaging (DTI)-MRI is an MRI technique that uses different gradient directions to generate images based on the random motion of water molecules. It is the most sensitive technique to depict microstructural changes in brain white matter. Secondary neurodegeneration shows a reduction in fractional anisotropy secondary to axonal and myelin sheath damage.”

2-Spelling and spelling errors need to be corrected.

Done.

3-In this section, more current literature information confirming the relevant data should be included, and this section should be shortened by summarizing it with the mentioned technical method.

Thanks for the suggestion. The content of this section was organized in a sequential manner by addressing the different techniques (T2W-MRI, DWI-MRI, DTI-MRI) in technological order of use and making the reader retrace the different information content resulting from technological advancement. At the same time we tried to keep the relationship between the signal alterations detected and the histopathology deriving from studies on the animal model and, to a limited extent, on humans clearly highlighted. The objective of the review is basically this and an excessive modification of this organization of the section would risk not fully achieving it. We transferred some sentences into a new table in order to shorten the section. Another aspect is that we are trying to balance the changes based on the suggestions of the different reviewers and the indications of the Editor. We are available to make further changes if deemed necessary by the Publisher.

  1. Regarding the 'Conclusion' section

1-The relevance of each technique included in this section to the subject of the article should be concluded. If there is a meaningful technique, it should be included with quantitative results.

Please, see the answer to a similar observation in the abstract section.

We added the following sentences in the conclusion section: “Several MRI techniques might be used to characterize this phenomenon and its course, but each of these tecnhiques has several limitations. T2W-MRI identifies Wallerian degeneration when the tract is completely disrupted after weeks from the injury. DWI-MRI al-lows for an early identification of Wallerian degeneration, but it takes few weeks to be ap-parent, Finally DTI-MRI could allow an early identification of Wallerian degeneration, but at the moment it provides relevant pathophysiological information without quantitative threshold helping to diagnose this phenomenon in individual patients in clinical practice.”

Reviewer 2 Report

Comments and Suggestions for Authors

The authors wrote a review article on the secondary neurodegeneration of the corticospinal tract primarily analyzing this pathology in the context of stroke. Among the post-stroke secondary neurodegeneration events, the authors focused on the neuroimaging features of the sub-type known as anterograde axonal (i.e., Wallerian) degeneration, which is a very debilitating condition as it directly affects motor functions. The authors discussed the mechanisms and the timing of the appearance of these pathological lesions in neuroimaging studies especially on some diffusion-based MRI techniques such as Diffusion Tensor Imaging (DTI), which may show alterations as early as within 7 days post-stroke. Finally, the authors discussed the potential importance of the identification of secondary neurodegeneration as a prognostic indicator for motor outcomes after stroke. The authors conclude that post-stroke secondary neurodegeneration in the CNS is a relatively neglected phenomenon that is insufficiently investigated. They also stress that a timely diagnosis in the early phase post-stroke could help design neuroprotective trials which could further help alter the motor outcomes in this patient population. Currently, there are no well-defined neuroimaging protocols (i.e., conventional MRI and DTI metrics) that can be used to extract meaningful prognostic information by leveraging these neuroimaging techniques. 

I found the manuscript to be well-written and quite informative for the potential reader as it brings into focus a type of neuropathology that is insufficiently investigated and could significantly benefit from additional clinical studies. My only comments are:

1.        I would recommend adding a few arrows to the individual images shown in Figure 1 to clearly indicate the location of the corticospinal tract lesions on these sequential slices chosen for exemplification.

2.      There are a number of typos throughout the manuscript that need to be fixed.    

Author Response

The authors wrote a review article on the secondary neurodegeneration of the corticospinal tract primarily analyzing this pathology in the context of stroke. Among the post-stroke secondary neurodegeneration events, the authors focused on the neuroimaging features of the sub-type known as anterograde axonal (i.e., Wallerian) degeneration, which is a very debilitating condition as it directly affects motor functions. The authors discussed the mechanisms and the timing of the appearance of these pathological lesions in neuroimaging studies especially on some diffusion-based MRI techniques such as Diffusion Tensor Imaging (DTI), which may show alterations as early as within 7 days post-stroke. Finally, the authors discussed the potential importance of the identification of secondary neurodegeneration as a prognostic indicator for motor outcomes after stroke. The authors conclude that post-stroke secondary neurodegeneration in the CNS is a relatively neglected phenomenon that is insufficiently investigated. They also stress that a timely diagnosis in the early phase post-stroke could help design neuroprotective trials which could further help alter the motor outcomes in this patient population. Currently, there are no well-defined neuroimaging protocols (i.e., conventional MRI and DTI metrics) that can be used to extract meaningful prognostic information by leveraging these neuroimaging techniques. 

I found the manuscript to be well-written and quite informative for the potential reader as it brings into focus a type of neuropathology that is insufficiently investigated and could significantly benefit from additional clinical studies. My only comments are:

  1. I would recommend adding a few arrows to the individual images shown in Figure 1 to clearly indicate the location of the corticospinal tract lesions on these sequential slices chosen for exemplification.

Many thanks! We added the arrows.

  1.      There are a number of typos throughout the manuscript that need to be fixed.  

Done.

Reviewer 3 Report

Comments and Suggestions for Authors

The topic of neuroimaging secondary neurodegeneration of the corticospinal tract is important, but the information presented in the manuscript is well known.

In its present form, the manuscript looks more like a chapter from a monograph, but not a narrative review. The use of outdated or well-known data for highly rated scientific journals is not recommended.

It is noteworthy that 52 of the 64 articles cited by the authors were published more than 10 or more years ago.

The tables contain well-known information that has been previously published, but there are no references to the authors. It is unclear how this review differs from the previously published ones.

Comments on the Quality of English Language

Minor correction

Author Response

We would thank the reviewer for his/her suggestions and comments. We already made some changes in the paper according with the suggestions from other reviewers in the attempt of balancing various opinions. 

The topic of neuroimaging secondary neurodegeneration of the corticospinal tract is important, but the information presented in the manuscript is well known.

You are right and we agree with you, but our aim is not to provide novelties because there are no news about the pathophysiology of this topic. Our aim was to sumamrize pathophysiology of SND (Wallerian degeneration in particular) in stroke and to focus on tissue changes underlying MRI findings in the various stages of Wallerian degeneration. 

In its present form, the manuscript looks more like a chapter from a monograph, but not a narrative review. The use of outdated or well-known data for highly rated scientific journals is not recommended.

It is noteworthy that 52 of the 64 articles cited by the authors were published more than 10 or more years ago.

The tables contain well-known information that has been previously published, but there are no references to the authors. It is unclear how this review differs from the previously published ones.

Once again, we do not declare to have new data to provide (in fact, ot is a narrative review). The manuscript has been designed and drafted coherently with the aims we declared.  Moreover, old papers are often neglected, but, in this case (and in other conditions/diesease in cerebrovascular field, where embryiological information might underlie the identification and understanding of vascular anomalies) they are fundamental and it would be an error to forget them and to consider more recent papers are better, tout court. The lack of recent papers derives from the lack of attention to this phenomenon. 

Our objective is to draw attention to a phenomenon that is currently neglected, but still exists and deserves mention and knowledge, even if only so as not to forget (for the new generations) what has been defined and known in the past.

Round 2

Reviewer 1 Report

Comments and Suggestions for Authors

Major changes suggested in the article have been implemented.

Author Response

Many thanks!

Reviewer 3 Report

Comments and Suggestions for Authors

I thank the authors for responding to my comments.

The manuscript has been modified, but the fundamental remarks, unfortunately, have not been eliminated. Still, there are no links in several tables.

Recent publications have not been analyzed.

See: https://towson.libguides.com/expert-reviews/narrative-literature-reviews.

Despite the relevance of this narrative review, its novelty and contribution to the field of research are still small.

Author Response

We understand the issues raised by the reviewed, as previously said, but our proposal was clearly and honestly presented, starting from the abstract and the introduction in its aims, i.e. to underline the pathological changes responsible for MRI signal in the different stages of Wallerian degeneration. It is not a review about secondary degeneration or post-stroke secondary degeneration in general. The topic is not new and the main literature is relaltively old, but it still provides the  most reliable information to pursue our aim. Recent papers are mainly focused on clinical information (PMID: 33253986) or are pictorial reviews (PMID: 34712899), case reports. etc. They are not suitable for our aim and they do not provide new relevant information about it. 

We added the references in table 2 and 3. 

This is not the fisrt narrative review we drafted for MDPI journals and we followed the journal's instructions and the evaluation of papers already published, including on other topics, by other authors.

We regret that we were unable to make all the requested changes, but our motivations were clearly expressed and the paper met with the approval of the other reviewers.